# In Vitro and In Silico Antistaphylococcal Activity of Indole Alkaloids Isolated from *Tabernaemontana cymosa* Jacq (Apocynaceae)

Yina Pájaro-González [1,2], Julián Cabrera-Barraza [1], Geraldine Martelo-Ramírez [1], Andrés F. Oliveros-Díaz [1], Juan Urrego-Álvarez [1], Wiston Quiñones-Fletcher [3] and Fredyc Díaz-Castillo [1,*]

1 Laboratory of Phytochemical and Pharmacological Researches, School of Pharmaceutical Sciences, University of Cartagena, Cartagena 130002, Colombia; yinapajaro@mail.uniatlantico.edu.co (Y.P.-G.); juancaba2507@gmail.com (J.C.-B.); geramartelo@gmail.com (G.M.-R.); aoliverosd@unicartagena.edu.co (A.F.O.-D.); jurregoa1@unicartagena.edu.co (J.U.-Á.)
2 Research Group in Healthcare Pharmacy and Pharmacology, Faculty of Chemistry and Pharmacy, University of Atlántico, Barranquilla 080003, Colombia
3 Organic Chemistry of Natural Products, University of Antioquia, Medellín 050001, Colombia; wiston.quinones@udea.edu.co
* Correspondence: fdiazc1@unicartagena.edu.co

**Abstract:** The species of the genus *Tabernaemontana* have a long tradition of use in different pathologies of infectious origins; the antibacterial, antifungal, and antiviral effects related to the control of the pathologies where the species of this genus are used, have been attributed to the indole monoterpene alkaloids, mainly those of the iboga type. There are more than 1000 alkaloids isolated from different species of *Tabernaemontana* and other genera of the Apocynaceae family, several of which lack studies related to antibacterial activity. In the present study, four monoterpene indole alkaloids were isolated from the seeds of the species *Tabernaemontana cymosa* Jacq, namely voacangine (1), voacangine-7-hydroxyindolenine (2), 3-oxovoacangine (3), and rupicoline (4), which were tested in an in vitro antibacterial activity study against the bacteria *S. aureus*, sensitive and resistant to methicillin, and classified by the World Health Organization as critical for the investigation of new antibiotics. Of the four alkaloids tested, only voacangine was active against *S. aureus*, with an MIC of 50 μg/mL. In addition, an in silico study was carried out between the four isolated alkaloids and some proteins of this bacterium, finding that voacangine also showed binding to proteins involved in cell wall synthesis, mainly PBP2 and PBP2a.

**Keywords:** monoterpene indol alkaloids; voacangine; PBP2a; methicillin-resistant *S. aureus*

## 1. Introduction

Among infectious diseases, bacterial causes represent one of the greatest current challenges for their treatment, due to the rapid appearance and dissemination of antibiotic resistance mechanisms, for which research into new antibacterial molecules is of vital interest and plants are one of the potential sources of them. Plants of the genus *Tabernaemontana*, one of the largest within the Apocynaceae family, with around 100 species distributed in the Neotropics and Paleotropics [1,2], have a tradition of use as anti-infectives, supported in the literature [3]. According to the review carried out by van Beek et al. (1984) [4], 30 species of *Tabernaemontana* have different uses that depend on their antimicrobial effect, for example, in warts, wounds, febrifuge, purgative, anthelmintic, syphilis, leprosy, ulcers and abscesses on the skin, dysentery, diarrhea, antiseptic, and elephantiasis. Other uses are also important, for example, in the inflammation of the nails, eyes, and throat, analgesic against headache, toothache and pain in general, and lastly, its action on the central nervous system, as a tonic and stimulant [3–8].

The antibacterial effect of the different *Tabernaemontana* species has been supported by in vitro studies against Gram-positive and Gram-negative bacteria and mycobacteria. The bacterium belonging to the Gram-positive group against which more studies have been carried out is *Staphylococcus aureus,* but there are also some studies against *Staphylococcus epidermidis*, *Enterococcus faecalis*, *Bacillus subtilis*, *Streptococcus pyogenes,* and *Streptococcus agalactiae*; and against some Gram-negative bacteria such as *Escherichia coli*, *Klebsiella pneumoniae*, *Pseudomonas aeruginosa*, *Salmonella enterica* subsp. enterica serovar Typhimurium, and *Shigella flexneri* [9–21].

The characteristic secondary metabolites of the plants corresponding to the genus *Tabernaemontana,* and to which the antimicrobial effect is attributed, are the monoterpenoid indole alkaloids [2]; more than 1800 structures of this type of alkaloids have been identified and grouped into three different classes (class I, class II, and class III) that include several types. These alkaloids are also found in other genera of the Apocynaceae family, such as *Voacanga*, *Tabernanthe*, *Tabernaemontana*, *Catharanthus*, *Coryanthe*, and *Aspidosperma* [2,22]. For some species of the genus *Tabernaemontana*, more than 100 structurally different alkaloids distributed in all the organs of the plant have been reported [1]. Despite the large number of monoterpenoid indole alkaloids isolated, most of them have not been evaluated against bacteria. Some of those that have been studied have minimum inhibitory concentration values below 10 µg/mL, such as the case of the alkaloids voacafricine A and B, isolated from *Voacanga africana*, which presented an MIC of 3.12 µg/mL against *S. aureus* and 0.78–6.25 µg/mL against *S. enterica* subsp. enterica serovar Typhi [23]. On the other hand, 3-hydroxy coronaridine, isolated from *Tabernaemontana glandulosa*, was very potent against *P. aureginosa* (MIC: 0.01µg/mL) [24]. Therefore, these results show the need to continue with the isolation and evaluation of this type of molecules from *Tabernaemontana* species.

According to the Catalog of Plants and Lichens of Colombia, 23 species of Tabernaemontana have been identified in this country, of which *Tabernaemontana cymosa* Jacq is the one with the greatest presence in the Colombian Caribbean region [25]. Of the alkaloids isolated from this species (coronaridine, voacangine-7α-hydroxyndolenine, voacangine, tabernosine, condylocarpine, 14,15-dehydro-16-epi-vincamine, heyneanine, tabernosine 3-oxo, 3-oxo voacangine, stemmadenine, stemmadenin-N-oxide, tabernosine-N-oxide, tetrahydroalstonin, and voacristine) [26], only the antibacterial activity of voacangine has been determined against *Mycobacterium tuberculosis*, *Mycobacterium avium* and *Mycobacterium kansaii* [27]. In our research group, *T. cymosa* has been found to be active against *Aedes aegypti* mosquito larvae [28] and against Dengue, Chikungunya, and Zika viruses [29,30], finding that those responsible for these effects correspond to the structures of the indole monoterpenoid alkaloids.

Although the antibacterial activity against methicillin-sensitive and resistant *S. aureus*, and the ethanolic extract of *T. cymosa* seeds at 512 µg/mL was null, in this research we carried out the in vitro study of four indole alkaloids (voacangine, voacangine-7-hidroxyindolenine, 3-oxovoacangine, and rupicoline) isolated from this ethanol extract. Additionally, we set out to perform an in silico assessment of the possible interactions among the alkaloids isolated from *T. cymosa* and the *S. aureus* proteins Acetylglucasamine-1-phosphate Uridyltransferase (GlmU), Aspartate Semialdehyde Dehydrogenase (ASADH), and Penicillin Binding Proteins (PBPs), as well as the ADME-Tox properties of this compound by applying oral bioavailability rules.

## 2. Materials and Methods

### 2.1. General Experimental Procedures

*Column chromatography* (CC): Silica gel 60 (70–230 mesh) was purchased from Merck Darmstadt, Germany; hexane (Hex), chloroform ($CHCl_3$), ethyl acetate (AcOEt), acetone ($Me_2CO$), and methanol (MeOH) analytical grade were purchased from Merck Darmstadt, Germany. *Thin layer chromatography* (TLC): Silica gel 60 F254 0.2 mm plates (Cat. 1.05729.0001) and Silica gel 60 F254 0.5 mm preparative plates (Cat. 1.05744.0001) were purchased from Merck Darmstadt, Germany. *Plate developing*: 1% vanillin (Merck, Darm-

stadt, Germany) in 10% $H_2SO_4$ (Merck Darmstadt, Germany) solution in ethanol; UV lamp 254 and 365 nm. *NMR*: [1]H, [13]C and 2D NMR spectra were recorded on a Bruker Fourier 300 spectrometer (Bruker Bio-Spin GmbH, Rheinstetten, Germany) operating at 300 MHz for [1]H and 75 MHz for [13]C NMR, using $CDCl_3$ (Sigma, St Louis, MO, USA) as the solvent, and TMS (Sigma, St Louis, MO, USA) as an internal standard. Chemical shifts (δ) are reported in ppm, and the coupling constants (J) are reported in Hz. *Antibacterial activity assays*: cation adjusted Mueller Hinton broth (CAMHB) and Mueller Hinton agar were purchased from Merck Darmstadt, Germany; dimethyl sulfoxide (DMSO) from Sigma, St Louis, Mo, USA; Vancomycin hydrochloride MP Biomedicals (LLC Cat No. 195540 Lot No QR14945); Oxacillin sodium salt, Sigma-Aldrich (28221-1G, Batch# 89022). *Optical density (OD) reading*: Skanlt 4.1/Multiskan FC, Thermo Scientific, Waltham, MA, USA, spectrophotometer.

### 2.2. Isolation and Characterization of T. cymosa Alkaloids

The seeds of *T. cymosa* were collected in the municipality of San Bernando del Viento located in the department of Cordoba, Colombia. The species was identified by the Guillermo Piñeres botanical garden scientist staff (Cartagena, Colombia, Voucher No. JBC 324). The seeds were extracted by maceration with 96% ethanol at a constant temperature of 25 °C. The extract was filtered out, then the ethanol was recovered in a rotary evaporator and reused in the maceration process. The dried extract (15 g) was fractionated using open column chromatography (5 × 60 cm) with Silica Gel; the fractionation was achieved using gradient as follows: dichloromethane (100%, 1 L), followed by dichloroethane–acetone (70:30, 1 L), dichloroethane–acetone (50:50, 1 L), acetone–methanol (50:50, 1 L), and methanol (100%, 1 L). Silica gel preparative plate chromatography (20 × 20 × 0.1 cm) was used to isolate the compounds. For the structural elucidation of the isolated compounds, the samples were analyzed in a Bruker Fourier spectrometer at 300 MHz for [1]H and 75 MHz for [13]C, obtaining one- and two-dimensional NMR spectral data; the solvent used was deuterated chloroform ($CDCl_3$) with tetramethyl-silane (TMS) as internal standard at a constant temperature of 25 °C.

### 2.3. Maintenance and Cultivation of Bacterial Strains

Two strains of *Staphylococcus aureus*, ATCC 29213 (methicillin sensitive) and ATCC 33591 (methicillin resistant), were kept in cryogenic vials at −80 °C. Once a month, a sample from the frozen strains was spread on Mueller Hinton agar and incubated at 37 °C for 24 h (first passage); subsequently, the plate was stored at 2–8 °C. One colony from the first passage was spread on Mueller Hinton agar and incubated at 37 °C for 24 h, the plate was stored at 2–8 °C, until one week (second passage). Overnight cultures in cation-adjusted Mueller Hinton broth (CAMHB) of some colonies from the second passage, were used to prepare the bacterial inoculum.

### 2.4. Minimum Inhibitory Concentration (MIC) of Indole Alkaloids

The minimum inhibitory concentration of the alkaloids voacangine, voacangine 7-hydroxyindolenine, 3-oxo-voacangine, and rupicoline was determined according to the protocol M07-11th edition of the Clinical and Laboratory Standards Institute (CLSI) [31]. A 10,000 µg/mL solution of each compound in dimethyl sulfoxide was prepared, from which an aliquot was taken and dissolved in CAMHB to obtain a final concentration of 128 µg/mL. Six serial dilutions were made in 96-well plates, the highest concentration being 128 µg/mL and the lowest 4 µg/mL (the range of concentrations finally evaluated was 2–64 µg/mL), leaving in each well a volume of 100 µL. Subsequently, 100 µL of a bacterial suspension (prepared from the overnight culture) were added to each dilution to achieve a quantity of bacteria equivalent to $5 \times 10^5$ CFU/mL. 1% dimethyl sulfoxide (DMSO) was used as a control for bacterial growth and the antibiotics oxacillin and vancomycin were used as a positive control for growth inhibition. The plates were incubated at 37 °C for 18 h and then the optical density in each well was read in a spectrophotometer at 620 nm.

The minimum inhibitory concentration of the evaluated compounds was determined using the optical density readings to calculate the percent of growth inhibition as previously published [32]. This experiment was done three times with three technical replicates.

### 2.5. In Silico Study
Ligands and Receptors Preparation

The structures of voacangine, voacangine hydroxyindolenine, rupicoline, and 3-oxovoacangine isolated from *T. cymosa* were obtained from the PubChem database [33] in SDF format; the subsequent geometric optimization was carried out with the Gaussian v.9 program [34] applying density functional theory as a method. The three-dimensional structures of the molecular targets corresponding to the penicillin-binding protein isotypes of *S. aureus* PBP1 (PDB ID: 5TRO), PBP2 (PDB ID: 3DWK), PBP3 (PDB ID: 3VSL), PBP4 (PDB ID: 5TXI), and PBP2a (PDB ID: 4CJN) were obtained from the Protein Data Bank (PDB) database as co-crystallized complexes with the exception of PBP1 [35]; on the other hand, the structure of GlmU was obtained from the UniProt database and unlike PBPs was determined by molecular modeling [36]. The structure of ASADH was constructed from the homology modeling technique using the SWISS MODEL program [37] from a FASTA sequence belonging to the resistant strain *Staphylococcus aureus* subsp. *aureus* MRSA252 downloaded from GenBank.

Geometric optimization of the proteins was performed using SYBYL-X 2.1.1. (SYBYL® from Tripos Inc., Princeton, NJ, USA) applying Kollman All-Atom force field, with a maximum of 1000 iterations and Amber type charges. In the structures of GlmU and ASADH, only their hydrogens were optimized, since they are molecular models. The preparation and determination of the grid box coordinates of each of the receptors was performed using the program AutoDock Vina 4.2.6., La Jolla, CA, USA [38] (https://vina.scripps.edu) (accessed on 24 April 2022) by adding polar hydrogens and Amber-type charges.

### 2.6. Molecular Docking between Indole Alkaloids from T. cymosa and S. aureus Proteins

The modeled structures were validated using SAVES server [39] and the protein-ligand docking method was validated prior to the assay by applying the pose selection method on the structures discharged in the form of co-crystallized complex, i.e., PBP2, PBP3, PBP4, and PBP2a. Then, molecular docking was performed in triplicate between the different molecular targets and voacangine employing AutoDock Vina 4.2.6., through which the different affinity values were obtained and the visualization of the presented interactions were obtained by the program Discovery Studio [40]. The binding site on each of the proteins was automatically determined by AutoDock Vina [38]. Additionally, prediction of the ADMET properties of the voacangine isolated from *T. cymosa* was performed through the FAF-Drugs 4 server [41,42].

### 2.7. Statistical Analyses

The mean of the percentages of growth inhibition of each treatment (alkaloids) and the control (antibiotics and DMSO) was used to calculate the MIC value.

## 3. Results
### 3.1. Isolation and Characterization of the Alkaloids Isolated from T. cymosa

Four indole alkaloid-type compounds from the seeds of *T. cymosa* (Figure 1), voacangine (**1**), voacangine-7-hydroxyindolenine (**2**), 3-oxovoacangine (**3**), and rupicoline (**4**), were isolated and submitted for structural characterization. The identification of the alkaloids isolated in this study was achieved by comparison of the NMR spectral data, mainly of the most representative $^{13}$C and $^{1}$H chemical shifts, with those of the literature. The isolated alkaloids in this work were reported in previous studies done by our group, where they showed activity against viruses of public health interest such as dengue and zika [30,43]. Table 1 shows the chemical shifts of $^{13}$C and $^{1}$H for each of the molecules. Spec-

troscopic data, chromatographic, and phytochemical behavior were extensively discussed and reported previously [29,30,44].

**Figure 1.** Chemical structures of indole alkaloids from *Tabernaemontana cymosa*.

**Table 1.** NMR Spectral data of indole alkaloids isolated from *T. cymosa*.

| Carbon | Voacangine | | Voacangine-7-hydroxyindolenine | | Rupicoline | | 3-Oxo-voacangine | |
|---|---|---|---|---|---|---|---|---|
| | $^{13}C$ | $^{1}H$ | $^{13}C$ | $^{1}H$ | $^{13}C$ | $^{1}H$ | $^{13}C$ | $^{1}H$ |
| 2 | 137.65 | | 186.97 | | 68.39 | | 134.65 | |
| 3 | 51.6 | | 48.75 | | 52.07 | | 173.07 | |
| 5 | 53.24 | | 49.22 | | 47.67 | | 42.72 | |
| 6 | 22.33 | | 34.25 | | 25.77 | | 21.18 | |
| 7 | 100.84 | | 88.43 | | 202.9 | | 109.22 | |
| 8 | 129.31 | | 144.53 | | 121.78 | | 128.23 | |
| 9 | 110.24 | 6.94 | 108.09 | 6.91 | 104.63 | 7.02 | 100.48 | 6.95 |
| 10 | 154.11 | | 159.23 | | 153.82 | | 154.21 | |
| 11 | 111.96 | 6.83 | 113.85 | 6.81 | 126.89 | 7.07 | 112.59 | 6.81 |
| 12 | 111.23 | 7.16 | 121.49 | 7.36 | 114.12 | 6.76 | 111.38 | 7.37 |
| 13 | 130.13 | | 144.9 | | 154.21 | | 130.83 | |
| 14 | 27.44 | | 27.08 | | 31.1 | | 35.97 | |
| 15 | 32.14 | | 32.14 | | 31.1 | | 31.03 | |
| 16 | 55.25 | | 55.88 | | 52.12 | | 56.11 | |
| 17 | 36.67 | | 34.63 | | 30.8 | | 35.49 | |
| 18 | 11.82 | 0.91 | 11.71 | 0.86 | 12.14 | 0.91 | 11.41 | 0.86 |
| 19 | 26.86 | | 26.62 | | 28.69 | | 27.67 | |
| 20 | 39.27 | | 37.68 | | 35.81 | | 38.21 | |
| 21 | 57.69 | 3.56 | 58.68 | | 52.05 | 3.95 | 56.02 | |
| 22 | 176.03 | | 174.05 | | 174.57 | | 175.6 | |
| $CO_2CH_3$ | 52.74 | 3.73 | 53.39 | 3.7 | 52.07 | 3.3 | 53.09 | 3.77 |
| $OCH_3$ | 56.15 | 3.87 | 55.88 | 3.81 | 55.9 | 3.76 | 55.6 | 3.88 |

*3.2. Antibacterial Activity of the Alkaloids Isolated from T. cymosa*

Table 2 shows the MICs against the two strains of *S. aureus* evaluated. As can be seen, voacangine was the only alkaloid that showed a growth inhibition greater than 90% ($MIC_{90}$) up to the maximum concentration evaluated (64 μg/mL). The exact value of the $MIC_{90}$ (50 μg/mL) was later determined between the concentration range of 32 to 64 μg/mL.

**Table 2.** MICs of indole alkaloids against sensitive (MSSA) and methicillin-resistant *Staphylococcus aureus* (MRSA).

| Isolated Compounds | Bacterial Stains Tested | |
| --- | --- | --- |
| | MSSA (ATCC 29213) | MRSA (ATCC 33591) |
| | MIC$_{90}$ (µg/mL) | |
| Voacangine | 50 | 50 |
| Voacangine-7-hydroxyindolenine | >64 | >64 |
| 3-Oxo-voacangine | >64 | >64 |
| Rupicoline | >64 | >64 |
| Oxacillin | <0.5 | 256 |
| Vancomycin | 1 | 1 |

### 3.3. Validation of the Docking Method

The integrity and identity of the proteins was maintained after geometric optimization since the RMSD (Root Mean Standard Deviation) values between each of the structures before and after the process was less than 1 Å [45]. On the other hand, the RMSD values obtained during validation by pose selection [46] indicated that the conformational similarity between the pose of co-crystallized ligand and the pose of the ligand determined in silico is acceptable, since they did not exceed 2Å, hence, the program used during docking is valid [47,48].

### 3.4. Molecular Docking Affinities of T. cymosa Alkaloids against S. aureus Proteins

The results of the molecular docking of *T. cymosa* alkaloids against the *S. aureus* proteins are shown in Figure 2. The voacangine–PBP2 complex showed the best affinity with a binding energy of −8.10 Kcal/mol and it is represented as the darkest point in the heat map, followed by voacangine–PBP2a with a binding energy of −7.70 Kcal/mol. Considering that voacangine was the only compound isolated from *T. cymosa* that had in vitro antibacterial activity and also presented the best affinity values, the results will be focused on voacangines interactions with proteins from *S. aureus*.

**Figure 2.** Heat-map of the molecular docking affinities value (kcal/mol) for *T. cymosa* alkaloids with *S. aureus* proteins.

### 3.5. Molecular Docking of Voacangine and Proteins from S. aureus

The results of the docking between voacangine and the proteins from *S. aureus* are shown in Table 3, where the binding energies with each of the macromolecules are shown, as well as the residues that participated and the type of interactions that occurred. Most of the interactions observed between voacangine and the proteins evaluated were of the conventional hydrogen bridge type, Alkyl and Pi-Alkyl.

**Table 3.** Molecular docking analysis between voacangine and proteins of *S. aureus* to be evaluated as potential molecular targets.

| Compound | Protein | Database ID | Specific Amino Acid Residues Involved | Type of Interaction |
|---|---|---|---|---|
| Voacangine | PBP1 | 5TRO | Asp257, Asn175, Phe176, Val250 | Conventional H-bridge, Pi-Anion, Alkyl, Pi-Alkyl, Pi-Alkyl |
| | PBP2 | 3DWK Q8KHY3 | Phe305, Ser510, Asn523, Ala524, Asn312, Asp541 | Conventional H-Bridge, Hydrogen–Carbon Bridge, Pi-Donor H-Bridge, Pi-Sigma, Alkyl, Pi-Alkyl |
| | PBP2a | 4CJN A0A0J9X1X5 | Gln396, Tyr499, Lys281, Gly282, Leu285, Leu286 | Conventional H-Bridge, Pi-Donor H-Bridge, Alkyl, Pi-Alkyl |
| | PBP3 | 3VSL | Lys326, Lys375, Asp378, Glu329, Leu365 | Conventional H-Bridge, Pi-Anion, Alkyl |
| | PBP4 | 5TXI | Arg280, Ile216, Lys217, Tyr374 | Conventional H-bridge, Alkyl, Pi-Alkyl |
| | GlmU | Q6GJH2 | Lys108, Arg242, Tyr246, Val266 | Conventional H-bridge, Hydrogen–Carbon Bridge, Pi-Pi T-Shaped, Alkyl |
| | ASADH | A0A7U7EUW2 | Asn94, Cys126, Val13, Leu12, Gly159 | Conventional H-bridge, Hydrogen–Carbon Bridge, Non-Favorable Donor-Donor, Alkyl, Pi-Alkyl |

Position and identification of the amino acid residues involved in each of the molecular dockings and the type of interaction carried out. H-bridge: Hydrogen bridge.

PBP2 is a bifunctional protein of *S. aureus* with a glycosyltransferase (GT) domain consisting of amino acids 93-290, Glu114 being the catalytic residue, and a transpeptidase (TP) domain consisting of amino acids 293-692 whose catalytic residue is Ser398 [49]. The interactions of voacangine and PBP2 docking occurred in the TP domain without coinciding with the catalytic residue. Figure 3 shows the binding mode of voacangine to PBP2, the complex that showed the best binding energy.

PBP2a is a class B transpeptidase associated with *S. aureus* resistance due to its low binding affinity to β-lactam antibiotics; this protein has two domains: the allosteric domain (amino acid 27-326) and the transpeptidase domain (amino acids 327-668), with Ser403 being the catalytic residue [50]. The binding site between voacangine and PBP2a was located in the allosteric domain of the above-mentioned protein, the binding mechanism is described in Figure 4.

On the other hand, voacangine showed acceptable affinity against PBP1, PBP3, PBP4, GlmU, and ASADH (see Figure 1). The voacangine binding site on PBP1 during docking occurred in the transpeptidase domain of PBP1 without matching the catalytic motif (see Figure S1). The catalytic motif of the transpeptidase domain of PBP1 from *S. aureus* (PDB ID: 5TRO) corresponds to amino acids Ser114-Thr315-Phe316-Lys317, determined by local alignment with PBP2x from *Streptococcus pneumoniae* (PDB ID: 1PYYY) [51] due to the absence of the crystallographic structure of PBP1 from *S. aureus* co-crystalized with a ligand; the BLAST alignment showed a similarity of 34.20% and the RMSD was 1.031Å when comparing the tertiary structures of both structures.

PBP3 has two domains, an N-terminal or non-enzymatic domain consisting of amino acids 66-303 and a C-terminal transpeptidase domain encompassing residues 351-656, with Ser392 being the catalytic amino acid [52]. Docking with voacangine occurred in the transpeptidase domain without coincidence with the catalytic residue, Figure S2 shows the binding mode of this docking. PBP4 is composed of two domains, an N-terminal one with transpeptidase function (residues 56-288) where the catalytic amino acid Ser75 is located and a second C-terminal domain of unknown function consisting of seven strands (residues 317-382) [53,54]. Figure S3 illustrates the interactions between voacangine and PBP4 of *S. aureus* at the binding site that was located in the transpeptidase domain without matching the catalytic residue. The GlmU structure (acetylglucosamine-1-phosphate uridyl-transferase) of *S. aureus* has an N-terminal pyrophosphorylase (PPase) domain (residues 4-277) and left-handed β-helix acetyltransferase (LβH) (residues 252-437) [55]. Voacangine binding occurred in the intermediate α-helical region (residues 228-251) located between

the two Ppase and LβH domains of GlmU, with intervention of one amino acid from the Ppase region: Lys108, and one amino acid from the LβH region: Val266; Figure S4 shows the binding mode between voacangine and GlmU. The aspartate β-semialdehyde dehydrogenase (ASADH) family has a highly conserved active site that includes the following amino acid motifs: GxxGxVG, GAA, and SGxG [56]; in the ASADH structure of *S. aureus* constructed in this study, these catalytic motifs correspond to residues Gly310-Ala312 and Ser156-Gly159 of the C-terminal region and Gly8-Gly14 of the N-terminal region. In the docking, voacangine interacts with, among others, the amino acids Val13 and Leu12 belonging to one of the catalytic motifs of this enzyme, which indicates that the binding occurs in the active site of the protein. Figure S5 shows the binding mode of voacangine with the ASADH structure.

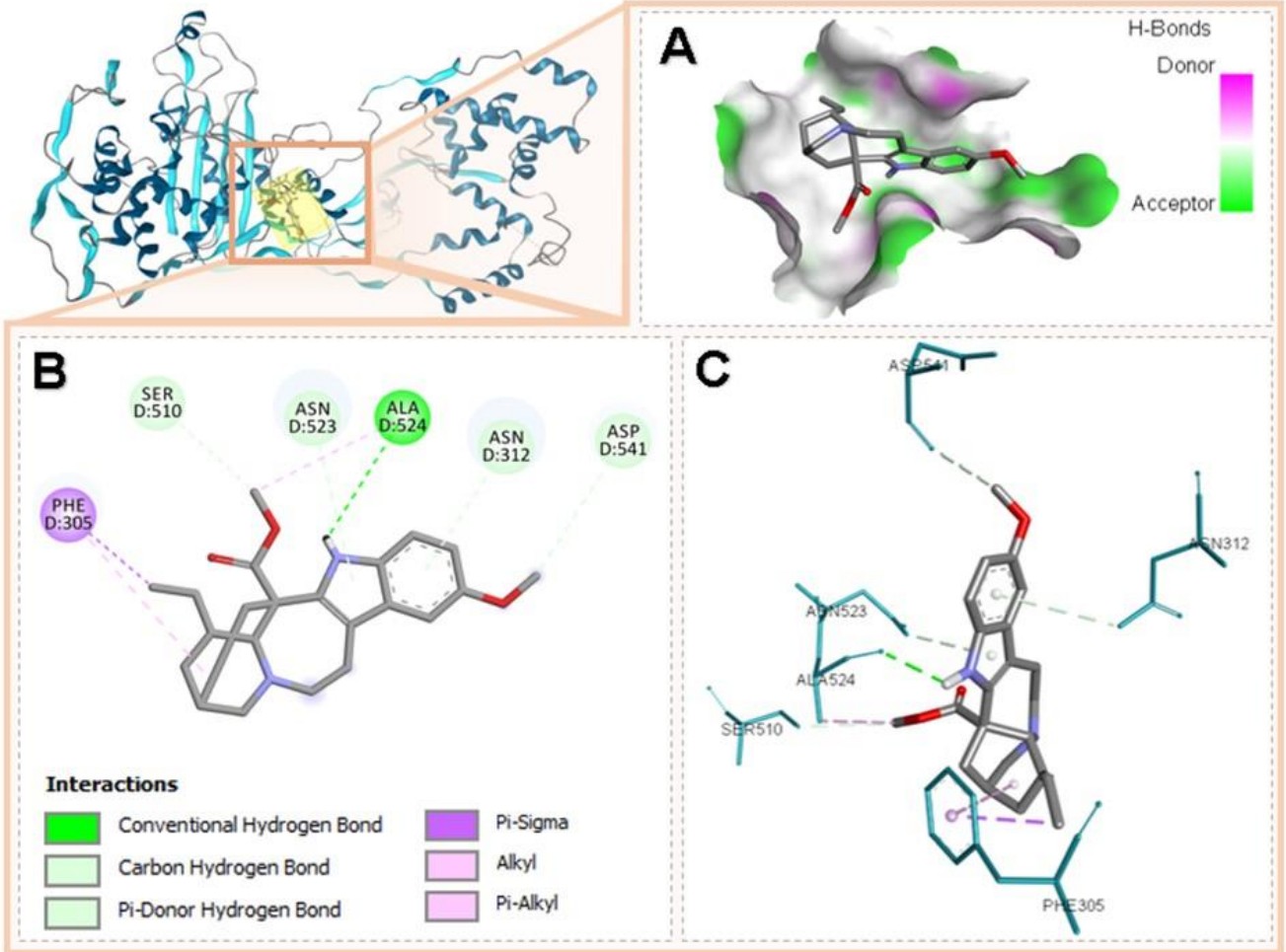

**Figure 3.** Molecular interactions between voacangine and PBP2 from *S. aureus.* (**A**) Characterization of the binding site according to the ability to form hydrogen bridges and geometry of the compound at the docking site, (**B**) 2D plot of the interactions specifying the type of interaction, (**C**) conformation of the compound during docking and interactions with PBP2 residues in 3D.

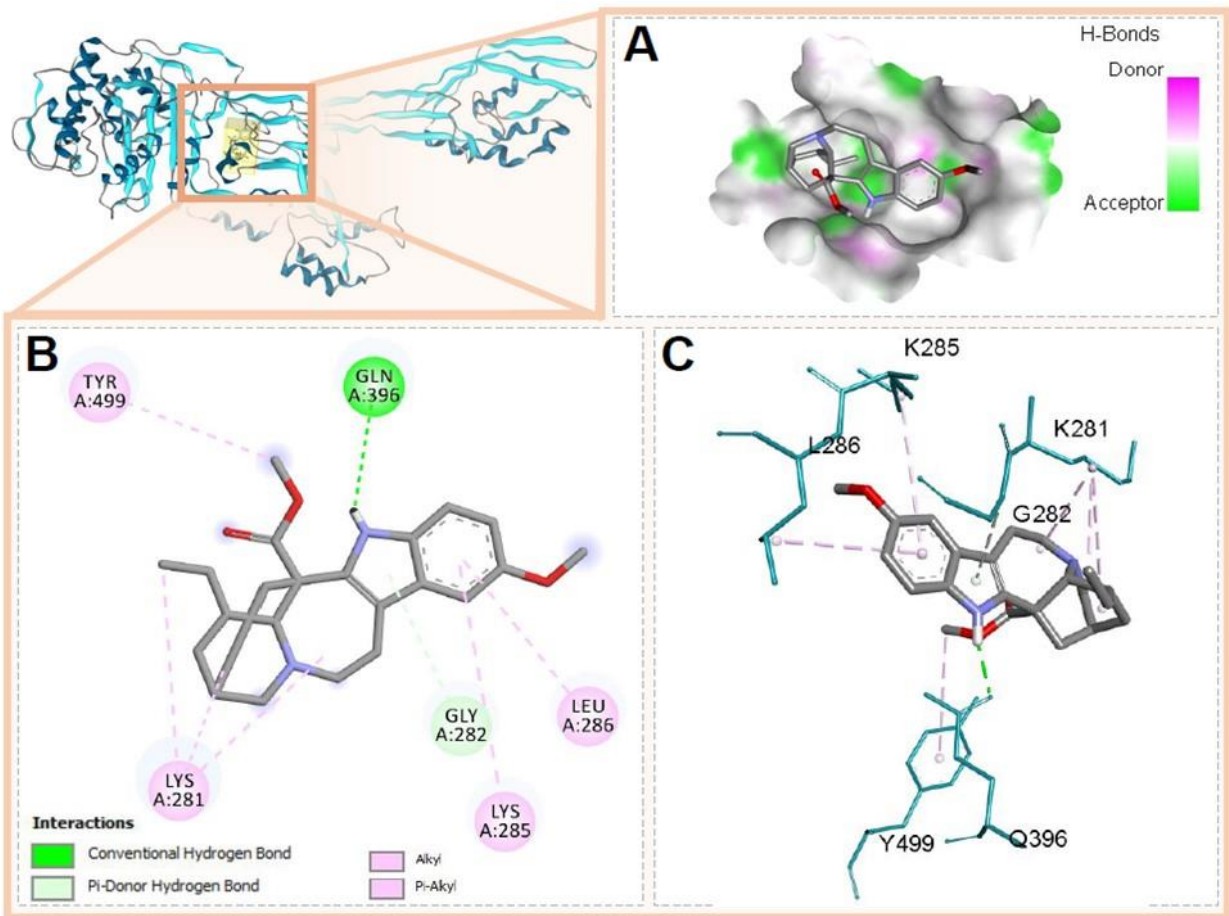

**Figure 4.** Molecular interactions between voacangine PBP2a and *S. aureus.* (**A**) Characterization of the binding site according to the ability to form hydrogen bridges and geometry of the compound at the docking site, (**B**) 2D plot of the interactions specifying the type of interaction, (**C**) conformation of the compound during docking and interactions with PBP2a residues in 3D.

### 3.6. ADMET Analysis of Voacangine Compound

In relation to the virtual analysis of ADMET properties (Table 4), it was determined that by applying Lipinski's rule of five [57–59] and Veber's rule [60], voacangine possesses the required conditions for good water solubility and intestinal permeability, essential characteristics in oral bioavailability. In addition, it is estimated to have low toxicity since no possible toxic groups were found in its structure when applying the FAF-Drugs evaluation [61–63], based on these results it can be stated that voacangine compound has potentially acceptable ADMET properties [64].

**Table 4.** Prediction of ADMET properties of voacangine using the FAS-Drugs server.

| Parameter Evaluated | Voacangine Compound |
|---|---|
| Molecular Weight | 368.47 |
| Log P | 3.51 |
| HBD | 1 |
| HBA | 5 |
| Rotating links | 4 |
| TPSA (Å$^2$) | 55.76 |
| Toxic functional groups | 0 |
| Result | Accepted |

Result of the parameters calculated by FAF-Drugs for each of the molecular structure of the voacangine compound isolated from *Tabernaemontana cymosa*.

## 4. Discussion

In a screening conducted by van Beek with 19 species of *Tabernaemontana*, nine extracts were active against the Gram-positive bacteria *B. subtilis* and *S. aureus* at 5 mg/mL and three extracts (*Tabernaemontana chippii*, *Tabernaemontana divaricata*, and *Tabernaemontana orientalis*) showed activity against a Gram-negative bacterium [3]. Most of the studies performed with extracts of *Tabernaemontana* species have shown a low to moderate activity, while others report a very high activity, for example, a fraction obtained from the ethanolic extract of *Tabernaemontana catharinensis* shows minimum inhibitory concentration values of <0.02 at 0.08 mg/mL (20–80 μg/mL), against strains of *S. aureus*, *S. epidermidis*, *E. coli* and *P. aeruginosa*, *Enterococcus* spp., *Klebsiella oxytoca*, *Citrobacter* spp., *K. pneumoniae*, and *Proteus mirabilis* [15]. One of the most active plants is *Tabernaemontana divaricata*, which presented a minimum inhibitory concentration (MIC) value of 4–128 μg/mL for Gram-positive bacteria and 16–128 μg/mL for Gram-negative bacteria [19].

The ethanolic extract of *Tabernaemontana cymosa* was inactive against *S. aureus* at a concentration of 512 μg/mL; however, the four isolated alkaloids were evaluated because no study related to their activity against this bacterium was found. Voacangine was the only alkaloid isolated from the seeds of *T. cymosa* that showed activity against methicillin sensitive and resistant *S. aureus* with an MIC of 50 μg/mL; being the same MIC value found against *Mycobacterium tuberculosis* in another study [27]; compared to the antibiotic oxacillin used as a control, voacangine was more active against the resistant strain of *S. aureus* (ATCC 33591). This MIC value is still too high to consider the molecule itself as an alternative to replace clinically used antibiotics, which have MIC values lower than 1 μg/mL; however, the results obtained from the in silico study performed in the present study indicated that voacangine might be required as a prototype for the development of new antistaphylococcal compounds.

The four alkaloids isolated from the seeds of *T. cymosa* in this study were evaluated in silico against seven proteins from *S. aureus*. As seen in the HeatMap of Figure 2, these four alkaloids showed a higher affinity for four of the proteins, namely PBP2, PBP2a, ASADH, and GlmU; PBP1, PBP3, and PBP4 proteins were not a target for these alkaloids. The in silico affinity order for voacangine was PBP2 > PBP2a > ASADH = GlmU; the greater affinity of voacangine for the PBP2 and PBP2a proteins could explain the result of in vitro growth inhibition of *S. aureus* sensitive and resistant to methicillin in this study. Voacangine's affinity for PBP2a is quite important in the control of methicillin-resistant strains.

The PBP2a protein has two domains, the allosteric domain (amino acids 27-326) and the transpeptidase domain (amino acids 327-668); this last domain is in such a conformation that the access of beta-lactam antibiotics and the endogenous ligand (peptidoglycan) to the Ser403 catalytic residue is limited [65,66]. A site in the allosteric domain, located 60Å from the catalytic site, is essential for the functioning of the enzyme as a mechanism of resistance to beta-lactam antibiotics; a peptidoglycan chain binds to this site, causing a conformational change in the catalytic site that allows the transpeptidation reaction to start; in this way, the site becomes available for its endogenous substrate (peptidoglycan) [67]. Knowledge of this resistance mechanism allows directing the search for antibacterial molecules that act at the level of the allosteric site and/or the catalytic site, as is the case with the anti-MRSA antibiotic ceftaroline; a molecule of this antibiotic mimics the union of a D-Ala-D-Ala residue of a nascent peptidoglycan chain, as a consequence the catalytic site of the enzyme is opened, on which another molecule of ceftaroline acts as an inhibitor [68–70].

As a result of the in silico study of voacangine against PBP2a, the alkaloid bound with higher affinity energy in the allosteric domain interacts with amino acids Gln396, Tyr499, Lys281, Gly282, Leu285, and Leu286. Members assigned to the allosteric site are found at the intersection of Lobe 1 (residues 166–240), Lobe 2 (residues 258–277), and Lobe 3 (residues 364–390). Although voacangine does not interact with any of them, their union could prevent the action of the endogenous ligand and therefore transpeptidation. According to the results, we suggest that the possible antibacterial mechanism of action of voacangine would be associated with the disruption of the peptidoglycan synthesis

pathway, either due to the inhibition of any of the PBP2 susceptible or resistant, or upstream of the signaling cascade of synthesis of peptiglycan, by inhibiting GlmU, which synthesizes UDP-GlcNAc preventing the entry of this last molecule into the process [71]. More studies are needed to confirm this hypothesis.

Voacangine has two advantages to be used as a starting point for the development of new anti-staphylococcal molecules; the first one is its yield, which is higher compared to other alkaloids of the same type in the genera *Tabernaemontana*, *Voacanga*, and *Tabernanthe*. For example, from the roots of the species *Voacanga africana*, voacangine could be obtained with a yield of 1.7%, compared to 0.3% for ibogaine [72]. In the course of our investigations with *T. cymosa*, we have calculated a voacangine yield of 1.5%, while for the other isolated alkaloids (3-oxo-voacangine, voacangine 7-hydroxyindolenine, and rupicoline) it was no greater than 0.05% based on the weight of seed extract. The second advantage stems from the result of the ADME-Tox analysis of the present study, in which voacangine showed adequate prediction of oral bioavailability. The two advantages mentioned above (performance and oral bioavailability) would allow the molecule to be selected for future antistaphylococcal drug development studies.

The estimation of the toxicity component in the ADME/tox voacangine analysis shows a low toxicity for the molecule. On the other hand, voacangine was evaluated in an in vivo toxicity assay against *C. elegans* up to 184 µg/mL showing a lethality of only 10% at this concentration (unpublished results). This is corroborated by the analysis of computational data recently published by our research group, where it was found that among ten indole alkaloid-type compounds identified in *T. cymosa* seeds extract, voacangine obtained the lowest binding energy values with 900 human protein complexes [44]. Of the most stable complexes between voacangine and human proteins, those formed with butyrylcholinesterase and Aldo keto reductase stand out with an average affinity of −8.9 Kcal/mol y −9.5 Kcal/mol, respectively, so this compound would be involved in processes related to the nervous system and oxidative stress. The above results suggest a possible selectivity between the antibacterial effect of the alkaloid against *S. aureus* compared to the toxic effects produced by this alkaloid in human cells and eukaryotic cells in general.

In our search for natural alternatives against multiresistant bacteria, we have found that the indole alkaloid voacangine isolated from the seeds of *T. cymosa*, against sensitive and resistant strains of *S. aureus*, showed a moderate effect with an MIC value of 50 µg/mL, fifty times less potent than the intrahospital antibiotic vancomycin. According to our results, voacangine formed very stable complexes with important proteins in *S. aureus* such as PBP2 and PBP2a (−8.1 and −7.7), which is directly related to the in vivo findings in sensitive and resistant strains, where the same inhibition concentration was found for both strains. The values of binding energies on PBP and GlmU proteins suggest a possible mechanism of action related to the interruption of cell wall synthesis.

## 5. Conclusions

In this research, the antibacterial activity against *S. aureus* of the indole alkaloids voacangine, 3-oxo-voacangine, voacangine 7-hydroxyindolenine, and rupicoline, isolated from the inactive ethanol extract of the seeds of *T. cymosa* (at a concentration of 512 µg/mL), was evaluated for the first time. Voacangine presented a very high MIC value (50 µg/mL) to be considered as a promising molecule for use as a clinical antibiotic. However, molecular docking studies, carried out by our group, demonstrated the binding of this alkaloid to PBP2a, indicating a possible mechanism of action, not only for this alkaloid, but also for other iboga-type indole alkaloids that are more active against *S. aureus*. Voacangine was obtained with high yields from its phytochemical isolation from the seeds of *T. cymosa* and also presented a low toxicity, demonstrated in the ADME/tox analysis and in the in vivo lethality study in *C. elegans*, a nematode used as an animal model that allows us to infer toxicity in humans. The in vitro and in silico studies carried out by our research group

showed that voacangine could be considered as a promising molecule for the hemisynthesis of more active and less toxic compounds.

**Supplementary Materials:** The following supporting information can be downloaded at: https://www.mdpi.com/article/10.3390/scipharm90020038/s1, Figure S1: Molecular interactions between PBP1 from *S. aureus* and voacangine. (A) Characterization of the binding site according to the ability to form hydrogen bridges and geometry of the compound at the docking site, (B) 2D plot of the interactions specifying the type of interaction, (C) conformation of the compound during docking and interactions with PBP1 residues in 3D; Figure S2: Molecular interactions between PBP3 from *S. aureus* and voacangine. (A) Characterization of the binding site according to the ability to form hydrogen bridges and geometry of the compound at the docking site, (B) 2D plot of the interactions specifying the type of interaction, (C) conformation of the compound during docking and interactions with PBP3 residues in 3D; Figure S3: Molecular interactions between PBP4 from *S. aureus* and voacangine. (A) Characterization of the binding site according to the ability to form hydrogen bridges and geometry of the compound at the docking site, (B) 2D plot of the interactions specifying the type of interaction, (C) conformation of the compound during docking and interactions with PBP4 residues in 3D; Figure S4: Molecular interactions between GlmU from *S. aureus* and voacangine. (A) Characterization of the binding site according to the ability to form hydrogen bridges and geometry of the compound at the docking site, (B) 2D plot of the interactions specifying the type of interaction, (C) conformation of the compound during docking and interactions with GlmU residues in 3D. Figure S5: Molecular interactions between ASADH from S. aureus and voacangine. (A) Characterization of the binding site according to the ability to form hydrogen bridges and geometry of the compound at the docking site, (B) 2D plot of the interactions specifying the type of interaction, (C) conformation of the compound during docking and interactions with ASADH residues in 3D.

**Author Contributions:** Conceptualization, Y.P.-G., A.F.O.-D., J.C.-B., J.U.-Á., G.M.-R. and F.D.-C.; methodology, Y.P.-G., A.F.O.-D., J.C.-B., J.U.-Á., W.Q.-F., G.M.-R. and F.D.-C.; investigation, Y.P.-G., A.F.O.-D., J.C.-B., J.U.-Á., G.M.-R., W.Q.-F. and F.D.-C.; resources, Y.P.-G., J.C.-B. and F.D.-C.; data curation, Y.P.-G., A.F.O.-D., J.C.-B., J.U.-Á., G.M.-R., W.Q.-F. and F.D.-C.; writing—original draft preparation, Y.P.-G., J.C.-B. and F.D.-C.; writing—review and editing, Y.P.-G., A.F.O.-D., J.C.-B., J.U.-Á., G.M.-R., W.Q.-F. and F.D.-C.; visualization, Y.P.-G., A.F.O.-D., J.C.-B., J.U.-Á. and F.D.-C.; supervision, Y.P.-G. and F.D.-C.; project administration, F.D.-C.; funding acquisition, Y.P.-G., J.C.-B. and F.D.-C. All authors have read and agreed to the published version of the manuscript.

**Funding:** This research was funded by Ministry of Science Technology and Innovation (Minciencias, Colombia), grant number 110777757752, Contract/Agreement No. 649-2018; University of Cartagena grant number 099-2018, Colombian National Doctorates Call No. 757/2016-Minciencias and Call N° 809/2018. The APC was funded by University of Cartagena.

**Institutional Review Board Statement:** Not applicable.

**Informed Consent Statement:** Not applicable.

**Data Availability Statement:** Not applicable.

**Acknowledgments:** The authors wish to thank the Universidad de Cartagena, Universidad del Atlántico, Universidad Metropolitana and for the financial support and the contribution of the facilities to carry out this research. They also thank the members of the research groups involved in this study.

**Conflicts of Interest:** The authors declare no conflict of interest. The funders had no role in the design of the study; in the collection, analyses, or interpretation of data; in the writing of the manuscript, or in the decision to publish the results.

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
