# Peer review of "In Vitro and In Silico Antistaphylococcal Activity of Indole Alkaloids Isolated from Tabernaemontana cymosa Jacq (Apocynaceae)"

_scipharm, doi:10.3390/scipharm90020038_

Round 1

Reviewer 1 Report

Dear Authors, 

The present study ID:scipharm-1742679 entitled "In vitro and in silico antistaphylococcal activity of indole alkaloids isolated from Tabernaemontana cymosa Jacq (Apocynaceae)" written by authors Yina Pajaro-Gonzalez, Julián Cabrera-Barraza, Geraldine Martelo-Ramírez, Andrés F. Oliveros Díaz, Juan Urrego-Álvarez, Wiston Quiñones-Fletcher, and Fredyc Díaz-Castillo.

The study is written on an interesting topic and is also current due to the need to find antimicrobial compounds. The study is generally prepared at an acceptable level, referring to fairly good literature.

However, I have the following comments on the study, which should be taken into account further in the text.

1/ Delete free lines in the introduction section - see L. 35, 45, etc.

2/ L. 47 - "in vitro" in italics

3/ unify writing "Gram-" vs. "gram-" (see L. 47, 48, etc.)

4/ L. 51 - the sentence should be rewritten, it does not give good meaning in English

5/ L. 52, etc. - Salmonella must be correctly identified

6/ L. 54 - italics

7/ Much of the text 46-63 should be moved to another place in the text (discussion, etc.)

8/ L. 138, etc. - methicillin

9/ MM section - missing manufacturers of materials

10/ L. 218-222, etc. - italics!

11/ concentration range better logically 2-64 ug / mL

12/ Conclusion is not written clearly, the text must include more conclusions and clear study results. 

Author Response

Response to Reviewer 1 Comments

Point 1: Delete free lines in the introduction section - see L. 35, 45, etc.

Response 1: According to the reviewer suggestion, we deleted free lines troughout the document.

Point 2: L. 47 - "in vitro" in italics

Response 2: According to the reviewer suggestion, we reviewed all the text and made corrections when “in vitro” were not in italic.

Point 3: unify writing "Gram-" vs. "gram-" (see L. 47, 48, etc.)

Response 3: According to the reviewer suggestion, we reviewed all the text and made corrections using only "Gram-negative" and "Gram-positive" troughout the document.

Point 4:  L. 51 - the sentence should be rewritten, it does not give good meaning in English

Response 4: We took into account the reviewer suggestion and rewrote the sentence.

Point 5:  L. 52, etc. - Salmonella must be correctly identified

Response 5: We took into account the reviewer suggestion and we identified Sallmonella correctly using Salmonella enterica subsp. enterica serovar Typhimurium.

Point 6:  L. 54 – italics

Response 6: According to the reviewer suggestion, we reviewed all the text and made corrections when the names of plants or bacteria were not in italic.

Point 7: Much of the text 46-63 should be moved to another place in the text (discussion, etc.)

Response 7: We took into account the reviewer suggestion and we moved the text from lines 52-63 to lines 319-330 (Discussion).

Point 8: L. 138, etc. – methicillin

Response 8: According to the reviewer suggestion, we reviewed all the text and made corrections using methicillin instead of meticillin.

Point 9: MM section - missing manufacturers of materials

Response 9: According to the reviewer suggestion, we reviewed section materials and methods and made corrections writing all the missing manufacturers of materials, lines 96-112.

Point 10: L. 218-222, etc. - italics!

Response 10: According to the reviewer suggestion, we reviewed all the text and made corrections when the names of plants or bacteria were not in italic.

Point 11: concentration range better logically 2-64 ug / mL

Response 11: According to the reviewer suggestion, we made the correction, line 145-146.

Point 12: Conclusion is not written clearly, the text must include more conclusions and clear study results. 

Response 12: According to the reviewer suggestion, we improved the redaction of conclusions as shown in lines 415-428.

Reviewer 2 Report

The article "In vitro and in silico antistaphylococcal activity of indole alkaloids isolated from Tabernaemontana cymosa Jacq (Apocynaceae)" describes the antibacterial activity of some substances isolated from T. Cymosa and the computed models for this properties.

The research reported in this paper is a part of the international effort of obtaining better materials with desired antimicrobial properties, which can help fight the bacterial resistance to antibiotics. The subject is worthy of investigation, the manuscript is generally correct, but the authors should made the indicated corrections.

Use Gram-positive (caps) consistently across the manuscript (Gram is a name) – see rows 54, 63.

Staphylococcus aureus should be in italics even in table legend (row 222). Same at rows 234 (also T. Cymosa) and 240.

Conclusion section must be reworked to underline the novelty and advantages of this research, with actual findings and values.

Author Response

Response to Reviewer 2 Comments

Point 1: Use Gram-positive (caps) consistently across the manuscript (Gram is a name) – see rows 54, 63.

Response 1: According to the reviewer suggestion, we reviewed all the text and made corrections using only "Gram-negative" and "Gram-positive" troughout the document.

Point 2: Staphylococcus aureus should be in italics even in table legend (row 222). Same at rows 234 (also T. Cymosa) and 240.

Response 2: According to the reviewer suggestion, we reviewed all the text and made corrections when the names of plants or bacteria were not in italic.

Point 3: Conclusion section must be reworked to underline the novelty and advantages of this research, with actual findings and values.

Response 3: According to the reviewer suggestion, we improved the redaction of conclusions as shown in lines 415-428.

Round 2

Reviewer 1 Report

The suggestions have been addressed. 

I recomend to publish the manuscript in Sci Pharm/MDPI.